# Inflection-Point Nutrition Support Determined by Oral Mucosal Apoptosis Rate Is a Novel Assessment Strategy for Personalized Nutrition: A Prospective Cohort Study

**DOI:** 10.3390/jpm12030358

**Published:** 2022-02-26

**Authors:** Chun Gao, Zike Li, Sheng Zhang, Dengyi Cao, Yang Yu, Yujie Zhang, Hao Chen, Dehua Fu, Jianping Gong

**Affiliations:** Department of Gastrointestinal Surgery, Tongji Hospital, Tongji Medical College, Huazhong University of Science and Technology, Wuhan 430000, China; gaochun75@hotmail.com (C.G.); d202181987@hust.edu.cn (Z.L.); aloof3737@126.com (S.Z.); caodengyi1986@126.com (D.C.); topyy@aliyun.com (Y.Y.); yujiezhang@outlook.com (Y.Z.); chenyisheng2020@163.com (H.C.); fdh1257919297@163.com (D.F.)

**Keywords:** inflection-point nutrition, gastric cancer, apoptosis, nutritional support, postoperative recovery

## Abstract

Background: Energy intake and nutritional status influences a patient’s recovery from major abdominal surgery. The aim of this study is to explore and validate the clinical feasibility of an inflection-point nutrition strategy for personalized nutrition in gastric cancer patients after surgery. Methods: We conducted a prospective cohort study from a single tertiary referral hospital. Patients diagnosed with gastric cancer who met the inclusion criteria were included in this study. We collected the demographic and clinic pathological characteristics of included patients. Patients were divided into a formular nutrition (FN) and inflection-point nutrition (IPN) group. We monitored the perioperative dynamics of the oral mucosal epithelia cell apoptosis rate. Predictive factors for inflection phenomenon were investigated in univariate and multivariate analysis. Results: A total of 53 gastric cancer patients were included. A total of 30 (56.6%) patients showed the inflection phenomenon, with 9 (34.6%) patients in the FN group and 21 (77.8%) patients in the IPN group, respectively. We found that patients with the inflection phenomenon had a shorter duration of hospital stay compared to patients without the inflection phenomenon (*p* = 0.04). In multivariate analysis, independent predictive factors for inflection phenomenon were age (*p* = 0.015), operation time ≤ 300 min (*p* = 0.012), and average energy intake ≥ 25 Kcal/kg/day (*p* = 0.038). Conclusions: Our findings for the first time revealed that the oral epithelial cell apoptosis rate can promptly reflect the patients’ perioperative nutrition needs. Meanwhile, we developing a novel and feasible nutrition therapy guided by the oral epithelial cell apoptosis rate is novel in gastric cancer patients that have undergone laparoscopic gastrectomy.

## 1. Introduction

Gastric cancer (GC) is a common malignancy in the digestive system and a leading cause of cancer death [1]. Surgery and perioperative chemotherapy were standard treatments for advanced gastric cancer [2,3,4]. Postoperative complications after gastrectomy with lymphadenectomy was associated with heavy morbidity and mortality in gastric cancer patients [5]. Postoperative complications may result in prolonged hospital stay, delayed postoperative chemotherapy, and an increased economic and psychological burden. Therefore, clinicians have conducted studies to investigate the risk factors for postoperative complications and relevant interventions.

Risk factors such as old age, malnutrition, comorbidities, and impaired immune function may relate to postoperative complications and delayed recovery from surgery [6,7,8]. Perioperative nutritional status has an important impact on the surgical outcome. Malnutrition and insufficient nutrition intake may lead to increased morbidity and mortality after major abdominal surgery [9,10]. Recent advances in perioperative management as enhanced recovery after surgery (ERAS) have had a profound contribution to personalized treatment and improved recovery after a major surgery [11,12]. However, traditional nutrition assessment methods are not accurate in precisely evaluating patients’ nutritional needs. In addition, a physical measurement of body composition and monitoring of serum proteins fail to promptly reflect pathophysiological nutritional changes in patients [13,14,15]. Therefore, clinicians have been investigating a novel strategy that can precisely reflect the perioperative nutrition status and energy needs of the patients.

In our previous study, we found that the oral mucosal cell apoptosis rate significantly correlated with the nutritional status in gastric cancer patients. Malnourished patients demonstrated a significant decreased apoptosis rate of the oral mucosal cell compared to patients with a normal nutrition status [16]. Measuring the oral mucosal apoptosis rate was a non-invasive technique and could be a precise nutritional assessment method. In this context, we aim to evaluate the value of the oral epithelial cell apoptosis rate detection in predicting patients’ perioperative nutrition needs. Furthermore, we attempt to establish a clinical nutrition therapy strategy guided by oral epithelial cell apoptosis rate detection in gastric cancer patients that have undergone laparoscopic gastrectomy with D2 lymphadenectomy.

## 2. Methods

### 2.1. Study Design and Participants

We performed a prospective cohort study in a single institution from April 2019 to December 2020. In total, 53 gastric cancer patients that underwent laparoscopic gastrectomy with D2 lymphadenectomy according to the Japanese Gastric Cancer Treatment Guidelines [17] were included in our study. The inclusion criteria were: (1) Patients with no nutritional risk (screened by NRS-2002); (2) patients with a histopathological confirmed diagnosis of gastric cancer and had undergone operation as laparoscopic gastrectomy with D2 lymph node dissection; (3) no previous history of malignancies or chemotherapy; (4) an absence of diabetes mellitus, hyperthyroidism, or other metabolic diseases; and (5) having no lesions of oral mucosa. Exclusion criteria included: (1) Intraoperative conversion to palliative tumor resection; (2) patients with a poor general status (Karnofsky < 80, Eastern Cooperative Oncology Group (ECOG) > 1); and (3) patients with a recent history of severe heart, lung, kidney, or liver failure.

This study was approved by the institutional medical ethics committee (TJ-20130803) and registered at clinicaltrials.gov under identification code ID: NCT02102659 and trial name: Biological Assessment of Clinical Nutrition and Its Application. This study was carried out in accordance with all aspects complying with the 1964 Helsinki Declaration and later versions. All participants provided their written informed consent.

### 2.2. Allocation of Patients

We applied a random block method with a 1:1 allocation for randomization using SAS software (version 9.4, SAS Institute Inc., Cary, NC, USA). A total of 169 patients were recruited for assessment. After the initial evaluation, 116 patients were excluded due to their poor general status, history of other malignancies, chemotherapy, or severe comorbidities. Finally, 53 eligible patients were randomly assigned to two groups: The formular nutrition (FN) group (*n* = 26) and inflection-point nutrition (IPN) group (*n* = 27). This study was conducted and presented in compliance with the Strengthening the Reporting of Cohort Studies in Surgery (STROCSS) guideline [18] (Appendix A). Figure 1 shows the flow diagram of participants through the study.

### 2.3. Monitoring of Apoptosis Rate of EOMECs

In all included patients, the apoptosis rate of exfoliated oral mucosal epithelial cells (EOMECs) was monitored from pre-operative day 1 to post-operative day 7 by employing the following methods: In brief, at early morning, after rinsing mouth with saline, the mucosal epithelial cells of patients were collected through scrubbing their mouths with a sterile cotton swab several times. The swab was then put into a sputum culture cup containing 15 mL 0.9% normal saline. After slight shaking, the solution was filtered through a 500-mesh sieve (BD Falcon) and centrifuged at 800 r/min for 5 min. The sediment was fixed with 80% ethanol and stored in a refrigerator at −20 °C for later use. Within 12 h, the cell suspension was taken out, shaken on a shaker for 10 s, and filtered through a 100-μm sieve (BD Falcon). Afterwards, the cell suspension was collected and centrifuged at 1200 r/min for 5 min. After removal of alcohol and washing with PBS, the sample was again centrifuged at 1200 r/min for 5 min. A total of 500 mL of 0.2 mol/L phosphate-citrate (PC) buffer (96 parts of 0.2 M Na_2_HPO_4_ and 4 parts of 0.1 M citric acid, pH 7.8) was added into the cell suspension. After shaking, the sample was stored at room temperature for 30 min, washed twice with PBS, and centrifuged at 1200 r/m for 5 min. After the addition of 5 mg/mL RNase 10 μL, 10 × PI (propidium iodide) 10 μL, and 180 μL PBS, the sample was stored at 4 °C in darkness, overnight and then analyzed by FACSort flow cytometry (BectonDickinson, San Jose, CA, USA) [19].

### 2.4. Clinical Management and Nutritional Support

Both patients in the FN and IPN group received intravenous nutrition support from postoperative day 1 to day 7. The total non-protein nutrition supplement included glucose and lipid (ratio 3:1), and electrolyte, micro-element, and vitamins according to the nutritional status of subjects and followed the standard procedure of the hospital. The doses of nutritional components were selected according to the Chinese Society for Parenteral and Enteral Nutrition (CSPEN) guideline on parenteral nutrition for patients undergoing surgery [20].

In the FN group, the daily non-protein energy need was calculated as follows: Patients were given the basal energy expenditure (BEE) on postoperative day 1 (BEE was calculated with the Harris–Benedict equations (for males: BEE = (66.47 + 13.75 × Weight (kg) + 5.00 × Height (cm) − 6.78 × Age (Y)) × 0.9; for females: BEE = [655.10 + 9.56 × Weight (kg) + 1.85 × Height (cm) − 4.68 × Age (Y)] × 0.9) [21,22]. From post-operative day 2, total daily energy expenditure (TDEE), after correction, was administered. In this study, the correction coefficient was empirically determined as 1.2. (Corrected equations: for males: TDEE = [66.47 + 13.75 × Weight (kg) + 5.00 × Height (cm) − 6.78 × Age (Y)] × 1.2; for females: TDEE = (655.10 + 9.56 × Weight (kg) + 1.85 × Height (cm) − 4.68 × Age (Y)) × 1.2.) The corrected TDEE was given until discharge. In the IPN group, the patients were given BEE on post-operative day 1, and corrected TDEE was administered on post-operative day 2. Thereafter, the non-protein calorie supply was increased by about 2 Kcal/(kg·d) until an inflection point occurred on the curve of the apoptosis rate of EOMECs or symptoms of nutrition intolerance developed. The amount of nutrition supply at the inflection point was maintained till discharged. The total daily energy supply in the IPN group was adjusted according to the mucosal apoptosis rate.

The primary endpoint of this study was to evaluate the impact of the inflection-point nutrition supply on postoperative hospital stay (the number of days from the day of operation until the date of discharge) and complication (according to the Clavien–Dindo criteria [23]). The secondary endpoints include the value of the oral epithelial cell apoptosis rate detection in predicting patients’ perioperative nutrition needs and the analyses of factors associated with the inflection phenomenon in gastric cancer patients after gastrectomy. Additionally, we recorded the following demographic and clinical data: 1. Demographic characteristics such as gender, age, body mass index (BMI), history of smoking or alcohol, and comorbidities; 2. laboratory characteristics such as complete blood count, total lymphocyte count, hemoglobin, albumin, prealbumin, serum tumor markers as carcinoembryonic antigen (CEA), and liver and kidney function tests; 3. clinical characteristics such as operation type, intra-operative blood loss, operation time, tumor invasion depth (T), presence of lymph node metastases (N), and tumor-node-metastasis stage (TNM); and 4. postoperative events such as first flatus and first defecation.

### 2.5. Statistical Analyses

We present categorical variables as whole numbers and percentages. Pearson chi-square test or Fisher exact tests were applied to compare the differences. We present continuous variables as the mean (standard deviation, SD)/medians (range) and analyzed with the student *t*-test or Mann–Whitney U test. The cut-off value for operation time and average energy intake were determined by the receiver operating characteristic (ROC) curve. The univariate logistic regression was used to evaluate potential predictive factors for inflection phenomenon. Only factors with *p*-value < 0.1 in univariate analysis were included in the final multivariate analysis model. Multivariate logistic regression was employed to identify independent predictive factors for inflection phenomenon. All *p* values were reported as two-sided with a significance level of 0.05. All statistical tests were performed in SPSS version 24.0 (IBM, Armonk, NY, USA) and graphing were performed by Graph Pad Prism version 8.00 software.

## 3. Results

### 3.1. Demographic and Clinical Characteristics of Included Patients

Between April 2019 to December 2020, 169 gastric cancer patients that had undergone laparoscopic gastrectomy with D2 lymph node dissection were initially selected for this study. After screening based on inclusion and exclusion criteria, 53 gastric cancer patients were included in final study (Figure 1). The main demographic, biochemical, and clinical parameters are summarized in Table 1 and Table 2. In general, the mean age of included patients was 56.2 ± 10.7 with 28 (52.8%) men and 25 (47.2%) women. A total of 31 (58.5%) and 38 (71.7%) had a history of smoking and alcohol, respectively. A total of 33 (62.3%) patients underwent laparoscopic distal gastrectomy, and 20 (37.3%) patients underwent laparoscopic total gastrectomy. The median duration of postoperative hospital stay was 11 days (IQR: 10–12). A total of 7 (13.2%) patients suffered from Grade 1 & 2 post-operative complications. No in-hospital death was reported in this study.

### 3.2. Prevalence and Description of Inflection-Point Phenomenon

In the inflection-point nutrition group, we found that as the nutrition supply increased, the apoptosis rate of oral mucosal epithelial cells rose to a certain point (the inflection point), and then entered a relative stable “plateau phase” (independent of further increase in nutrition intake). The above phenomenon was defined as “inflection-point phenomenon”. We defined the inflection point as the point that the oral mucosa apoptosis rate does not increase with the increase of energy and fluctuates within 5% (Figure 2A). In the FN group, 9 (34.6%) patients experienced “inflection-point phenomenon” in the postoperative dynamic change in oral mucosal cell apoptosis. In the IPN group, 21 (77.8%) patients experienced an “inflection-point phenomenon” (Figure 2B). In total, 30 (56.6%) patients showed an inflection phenomenon and 23 (41.1%) patients did not experience an inflection phenomenon.

Figure 3 demonstrates the representative figures for the inflection-point phenomenon. In general, the average apoptosis rate of oral mucosal cells decreased after surgery. In most patients with the inflection/point phenomenon, as the energy supply increased, the apoptosis rate of oral mucosal cells reached the inflection point on postoperative day 2 to day 4 (21/30, 70%). Appendix A demonstrated the representative figures for patients without an inflection-point phenomenon.

### 3.3. Predictive Factors Associated with Inflection Phenomenon

Predictive factors for inflection-point phenomenon were identified from univariate analysis (Table 3). Univariate analysis revealed that age (>55 y vs. ≤55 y, hazard ratio (HR) = 3.75, *p* = 0.024), body mass index (≤24 kg/m^2^ vs. >24 kg/m^2^ HR = 4.18, *p* = 0.037), decreased serum albumin (<35 g/L, HR = 4.44, *p* = 0.04), operation time (≤300 min vs. >300 min, HR = 4.55, *p* = 0.018), and average energy intake (≥25 Kcal/kg/day vs.<25 Kcal/kg/day, HR = 5.96, *p* < 0.01) were related to inflection phenomenon (Table 3). Moreover, multivariate logistic regression analysis showed that age (>55 y), operation time (≤300 min), and average energy intake (≥25 Kcal/kg/day) were identified as independent predictive factors associated with inflection phenomenon, with the HR of 6.8 (*p* = 0.015, 95%CI 1.45–31.84), 7.69 (*p* = 0.012, 95%CI 1.59–33.33), and 5.32 (*p* = 0.038, 95%CI 1.10–25.79), respectively. The details of univariate and multivariate analysis are listed in Table 3.

### 3.4. Inflection-Point Nutrition Supply and Postoperative Recovery

Table 4 shows the postoperative characteristics in two groups. In general, the overall rate of postoperative complication was 13.2% (7/53). Postoperative complications were observed in 3 patients (11.1%) in the IFN group and 4 patients (15.4%) in the FN group (*p* = 0.65). The median times to postoperative first flatus were 60 h (IQR: 48, 72) after surgery in the IFN group and 68 h (IQR: 48, 80) after surgery (*p* = 0.56) in the FN group. There were no significant differences in total hospital stay and postoperative hospital stay between two groups (*p* = 0.71 and 0.98).

## 4. Discussion

The nutritional status influenced the short-term and long-term outcome in patients with gastric cancer. In this prospective cohort study, we investigated the value of oral epithelial cell apoptosis rate detection in predicting patients’ nutrition needs. For the first time we described the inflection-point phenomenon and applied it in a novel clinical nutrition therapy strategy. In our study, 30 (56.6%) patients experienced inflection-point phenomenon. We explored the potential predictive factors for the inflection phenomenon, revealing that age (>55 y), shorter operation time (≤300 min), and sufficient average energy intake (≥25 Kcal/kg/day) were independent predictive factors (Table 3). The inflection phenomenon may positively associate with postoperative recovery. Therefore, sufficient postoperative energy supply may be a benefit for enhanced recovery after surgery.

Trauma caused by the surgery procedures will lead to an activated systemic inflammatory response and increased metabolic demands. Therefore, adequate nutrition intake was essential for optimal postoperative recovery after major abdominal surgery. According to the recommendation from the European Society for Clinical Nutrition and Metabolism (ESPEN), cancer patients should receive 1.5 g protein/kg/day and 25–30 kcal/kg/day for energy intake after major abdominal surgery [24]. Malnutrition and acute muscle loss were risk factors for delayed postoperative recovery and postoperative complication after gastrectomy [25,26,27]. Deficiency in traditional nutritional evaluation and supply methods were major causes for inadequate postoperative nutrition intake. It has been reported that most patients that underwent major abdominal surgery did not consume adequate protein and energy after surgery [28]. Inadequate protein and energy intake may be correlated with poor surgical outcomes such as higher complication risk and longer hospital stay [29].

In this study, we developed a novel nutrition therapy (inflection-point nutrition) based on the apoptosis rate of patients’ mucosa epithelial cells. Accurate nutrition assessment is essential for effective nutrition support therapy. In our previous study, we found that the apoptosis/proliferation ratio (A/P) of exfoliated oral mucosal epithelial cells (EOMECs) varied and associated with the change of body nutritional status. The rate of both apoptosis and proliferation was reduced in patients with malnutrition [19]. Oral epithelial cells were cells with active renewal and a strong proliferative state. The apoptosis and proliferative rate of the oral epithelial cell can be affected by the change of nutrition status of the body. Some animal studies have found that malnutrition leads to a lower apoptosis rate [30] while others indicate that malnutrition is associated with a significant increase of spontaneously apoptotic cells in the thymus and spleen [31]. At this stage we were not able to explain the potential underlying mechanism of the inflection-point phenomenon. Further studies were needed to explore the biological association between oral epithelial cell apoptosis and nutritional status.

By active monitoring dynamics of patients’ mucosal epithelia cell apoptosis, we found that the apoptosis rate elevated after nutrition support therapy, reflecting the efficacy of nutrition support therapy (Figure 3). After the energy supply increase, the oral epithelial cell apoptosis rate of most participants increased. The energy supply calculated by a conventional method may be insufficient for this group of patients. An increase in energy supply by nutrition therapy, results as a further increase of oral epithelial cell apoptosis. Therefore, we, for the first time, reported the inflection-point phenomenon, indicating the actual optimal energy needs of the individual patient. Appling oral epithelial cell apoptosis rate detection could identify patients’ actual energy needs and evaluate the efficacy of nutrition therapy. In our preliminary study, no significant differences were found in short-term clinical outcomes in postoperative complication and hospital stay between the inflection-point nutrition (IFN) group and formular nutrition group (Table 4). More outcome measurements may be necessary to validate the clinical implication of inflection-point nutrition.

In our study, 30 (56.6%) patients showed the inflection phenomenon. We explored the predictive factors for the inflection phenomenon and found that older age, a longer operation time, and average energy intake ≥25 Kcal/kg/day correlate with inflection phenomenon. Elder patients and longer operation time may result in an increase demand of total energy expenditure. Although we included patients with no nutritional risk, elder patients experiencing a higher possibility of inflection phenomenon may indicate that monitoring of oral epithelial mucosal cell apoptosis was better in reflecting the actual energy needs compared to traditional methods. Patients in the IFN group received a higher average energy intake than patients in the FN group. A higher average energy intake (≥25 Kcal/kg/day) significantly associated with the inflection phenomenon, suggesting the insufficient energy supply calculated by a traditional formular nutrition method. Interestingly, the inflection phenomenon was absent in 23 (41.1%) patients. The potential reasons could be detection errors, an insufficient energy supply before inflection-point was reached, fluctuation of mucosal cell apoptosis due to traumatic stress, or unbalanced catabolism (Appendix A).

Patients with inflection phenomenon may have a better short-term outcome after surgery. In our study, we found that patients with inflection phenomenon had a shorter duration of hospital stay compared to patients without inflection phenomenon (10 days vs. 11 days, *p* = 0.04) (Table 2). Inflection phenomenon indicated that the energy supply could satisfy the actual need of the individual. Therefore, this finding suggested that the nutrition strategy that can satisfy the actual energy need of an individual would be a better nutrition delivery method. This might result in a better recovery for gastric cancer patients undergoing gastrectomy.

## 5. Clinical Implication and Limitations

This study provided novel observations and clinical feasibility of inflection-point nutrition guided by the oral mucosal cell apoptosis rate in gastric cancer patients after gastrectomy with D2 lymphadenectomy. We, for the first time, developed a nutrition protocol that can meet the actual nutrition demand of patients after surgery. Therefore, inflection-point nutrition can be conveniently applied in clinical practice for both short-term (post-operative trauma, chemotherapy, and rapid rehabilitation) and long-term (intestinal fistula, short-bowl syndrome) nutrition support therapies.

We acknowledge that the present study has several limitations. First, the sample size of our study is relatively small. With only 30 (56.6%) patients having demonstrated inflection point phenomenon, it is difficult to draw a solid conclusion with statistical significance. Therefore, the results in our study might only be considered as indicative. Second, the primary endpoints of this study were the duration of hospital stay and postoperative complications. The correlation between inflection-point nutrition and functional outcomes regarding postoperative recovery in gastric cancer patients was not clear. Third, the defects in the sampling and detection process of the mucosal epithelial cell results in a fluctuation of the apoptosis rate. To overcome these hurdles, randomized clinical trials with a larger sample size and more clinical-pathological measurements are needed to validate the findings of our study. Meanwhile, we need to develop a more stable and reliable method to detect the apoptosis rate of mucosal epithelia cells.

## 6. Conclusions

In summary, our study for the first time indicated that the oral epithelial cell apoptosis rate could promptly reflect patients’ perioperative nutrition needs. Second, inflection nutrition guided by the oral epithelial cell apoptosis rate is a novel and feasible nutrition therapy in gastric cancer patients that have undergone laparoscopic gastrectomy.

## Figures and Tables

**Figure 1 jpm-12-00358-f001:**
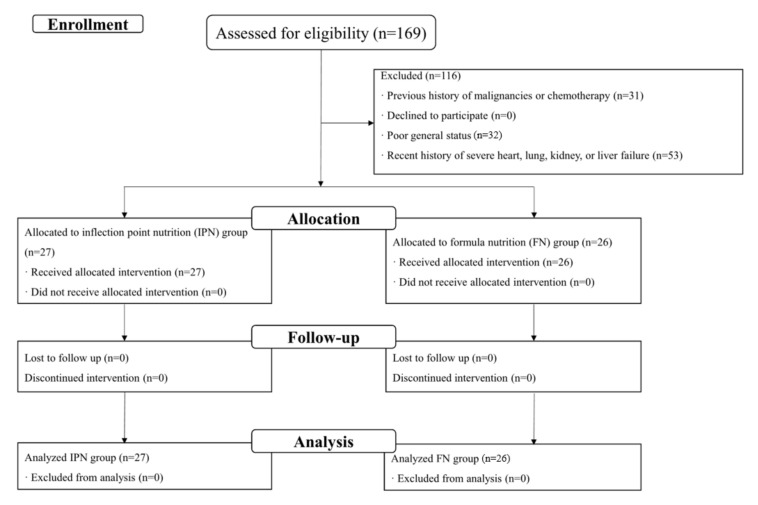
Flow diagram of the sequence through the stages of an observational.

**Figure 2 jpm-12-00358-f002:**
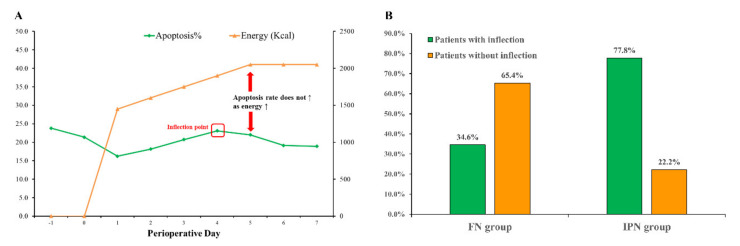
(**A**) Definition of inflection point (oral epithelial cell apoptosis rate does not increase with the increase of energy and fluctuates within 5%). (**B**) The percentage of patients experienced “inflection-point phenomenon” in the IPN group and FN group.

**Figure 3 jpm-12-00358-f003:**
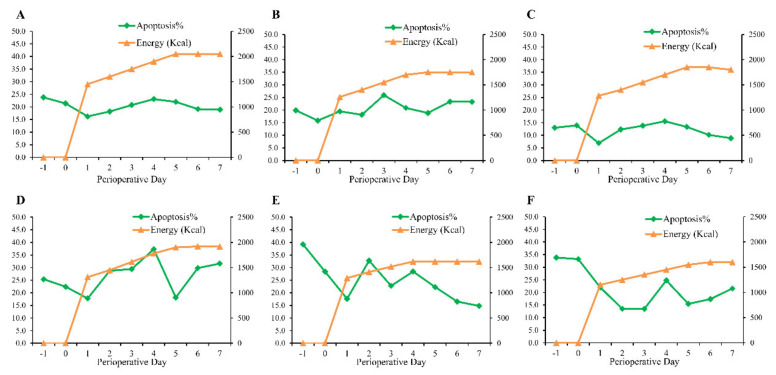
Representative figures for patients with inflection-point phenomenon. (**A**–**F**, 6 patients from the IPN group). *x* axis indicates perioperative day, *y* axis (left) indicates oral epithelial cell apoptosis rate (%), and *y* axis (right) indicates total energy intake (Kcal).

**Table 1 jpm-12-00358-t001:** The demographic and laboratory characteristics of all patients (*n* = 53).

Variables	All Patients (N = 53)	No inflection Phenomenon (*n* = 23)	Inflection Phenomenon (*n* = 30)	*p*-Value
**Demographic**				
Age, mean (SD), y	56.2 ± 10.7	53.7 ± 11.6	58.2 ± 9.7	0.130
Gender, Male/Female	28/25	12/11	16/14	0.933
Body mass index mean (SD), kg/m^2^	22.3 ± 2.4	22.9 ± 2.5	21.9 ± 2.2	0.141
Smoking				
No	58.5% (31/53)	56.5% (13/23)	60.0% (18/30)	0.799
Yes	41.5% (22/53)	43.5% (10/23)	40.0% (12/30)	
Alcohol				
No	71.7% (38/53)	73.9% (17/23)	70.0% (21/30)	0.754
Yes	28.3% (15/53)	26.1% (6/23)	30.0% (9/30)	
Comorbidity				
No	88.7% (47/53)	91.3% (21/23)	86.7% (26/30)	0.597
Yes	11.3% (6/53)	8.7% (15/23)	13.3% (4/30)	
**Laboratory**				
Albumin (g/L), mean (SD)	38.4 ± 3.9	39.3 ± 3.6	37.7 ± 4.1	0.144
Prealbumin (mg/L), mean (SD)	225.6 ± 44.2	214.1 ± 41.9	234.5 ± 44.7	0.095
Absolute neutrophil count (* 10^9^/L), mean (SD)	3.2 ± 1.1	3.2 ± 1.1	3.1 ± 1.2	0.780
Absolute lymphocyte count (* 10^9^/L), mean (SD)	1.9 ± 0.6	1.8 ± 0.6	1.9 ± 0.5	0.721
Hemoglobin (g/L), mean (SD)	122.4 ± 26.5	123.0 ± 24.5	121.7 ± 28.3	0.870
Preoperative serum CEA				
Normal (<5 ng/mL)	88.7% (47/53)	82.6% (19/23)	93.3% (28/30)	0.222
Elevated (≥5 ng/mL)	11.3% (6/53)	17.4% (4/23)	6.7% (2/30)	
Preoperative serum CA19-9				
Normal (<37 IU/mL)	96.2% (51/53)	95.7% (22/23)	96.7% (29/30)	0.848
Elevated (≥37 IU/mL)	3.8% (2/53)	4.3% (1/23)	3.3% (1/30)	

For *p*-value: The asterisk superscript indicates significant difference. SD, standard deviation; CEA, carcinoembryonic antigen.

**Table 2 jpm-12-00358-t002:** The perioperative characteristics of all patients (*n* = 53).

Variables	All Patients (N = 53)	No inflection Phenomenon (*n* = 23)	Inflection Phenomenon (*n* = 30)	*p*-Value
Blood loss, mL, mean (SD)	235 ± 114	235 ± 114	222 ± 133	0.76
Operation time, min, mean (SD)	324 ± 50	330 ± 45	318 ± 54	0.39
Operation				0.70
Distal gastrectomy	62.3% (33/53)	65.2% (15/23)	60.0% (18/30)	
Total gastrectomy	37.7% (20/53)	34.8% (8/23)	40.0% (12/30)	
Average energy intake, Kcal/Kg/day	27.4 ± 3.9	25.7 ± 3.1	28.8 ± 4.1	<0.01 *
TNM stage				
I/II	41.5% (21/53)	39.1% (9/23)	43.3% (13/30)	0.76
III/IV	58.5% (31/53)	60.9% (14/23)	56.7% (17/30)	
Median length of stay, day	11 (10–12)	11 (10–17)	10 (9–11)	0.04 *
Complications				
No	86.8% (46/53)	90.0% (27/340)	83.3% (19/23)	0.431
Grade 1 & 2	13.2% (7/53)	10.0% (3/30)	17.4% (4/23)	
Mortality	0 (0, 0)	0 (0, 0)	0 (0, 0)	1.00

For *p*-value: The asterisk superscript indicates significant difference. SD, standard deviation; TNM, tumor-node-metastasis; AC, adjuvant chemotherapy; RFS, recurrence-free survival.

**Table 3 jpm-12-00358-t003:** Univariate and multivariate logistic regression analyses of factors associated with the inflection phenomenon (*n* = 53).

		Inflection Phenomenon		
	Univariate		Multivariate	
Variable	HR (95%CI)	*p*	HR (95%CI)	*p*
Age				
≤55 y	1 (Ref)		1 (Ref)	
>55 y	3.75 (1.19–11.79)	0.024 *	6.80 (1.45–31.84)	0.015 *
Gender				
Male	1 (Ref)		N/A	
Female	1.05 (0.35–3.11)	0.93		
Body mass index, kg/m^2^				
>24	1 (Ref)		1 (Ref)	
≤24	4.18 (1.09–16.04)	0.037 *	5.83 (0.89–38.01)	0.065 *
Smoking				
No	1 (Ref)		N/A	
Yes	0.87 (0.29–2.61)	0.80		
Alcohol				
No	1 (Ref)		N/A	
Yes	1.21 (0.38–4.09)	0.75		
Comorbidity			N/A	
No	1 (Ref)			
Yes	1.62 (0.27–9.70)	0.60		
Albumin				
≥35 g/L	1 (Ref)		NS (*p* = 0.902)	
<35 g/L	4.44 (1.08–18.32)	0.04 *		
Prealbumin	1.01 (1.00–1.03)	0.105	N/A	
Hemoglobin	1.00 (0.98–1.02)	0.87	N/A	
TNM stage				
I/II	1 (Ref)		N/A	
III/IV	0.84 (0.28–2.54)	0.76		
Operation time				
>300 min	1 (Ref)		1 (Ref)	
≤300 min	4.55 (1.30–16.67)	0.018 *	7.69 (1.59–33.33)	0.012 *
Blood loss	1.00 (1.00–1.01)	0.76	N/A	
Operation				
Distal gastrectomy	1 (Ref)		N/A	
Total gastrectomy	1.25 (0.41–3.86)	0.70		
Average energy intake				
<25 Kcal/kg/day	1 (Ref)		1 (Ref)	
≥25 Kcal/kg/day	5.96 (1.57–22.60)	<0.01 *	5.32 (1.10–25.79)	0.038 *
Complication				
No	1 (Ref)		N/A	
Yes	0.53 (0.11–2.64)	0.44		

For *p*-value: The asterisk superscript indicates significant difference. HR, hazard ratio; TNM, tumor-node-metastasis; N/A, not applicable; NS, no significant difference.

**Table 4 jpm-12-00358-t004:** Postoperative recovery of patients received inflection-point nutrition (IPN) and formular nutrition (FN) (*n* = 53).

Variables	Inflection-Point Nutrition Group (*n* = 27)	Formular Nutrition Group (*n* = 26)	*p*-Value
First flatus, h, median (IQR)	60 (48, 72)	68 (48, 80)	0.56
Overall complications	3 (11.1%)	4 (15.4%)	0.65
Fever	0	1	
Pulmonary infection	0	1	
Small amount of pleural effusion	2	0	
Gastroparesis	1	0	
Wound problem	0	1	
Intraluminal bleeding	0	1	
Total hospital stay, days, median (IQR)	18 (16, 21)	17 (15, 24)	0.71
Postoperative hospital stay, days, median (IQR)	11 (10, 12)	11 (9, 13)	0.98

## Data Availability

The database used and/or analyzed during the current study is not publicly available (to maintain privacy) but can be available from the corresponding author on reasonable request.

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
