# Peer review of "Inflection-Point Nutrition Support Determined by Oral Mucosal Apoptosis Rate Is a Novel Assessment Strategy for Personalized Nutrition: A Prospective Cohort Study"

_jpm, 2022, doi:10.3390/jpm12030358_

Round 1

Reviewer 1 Report

In the manuscript titled Inflection-point nutrition support by oral mucosal apoptosis rate is novel assessment strategy for personalized nutrition: A prospective cohort study, Gao C. et al.

presented the results of studies which show an association between the rate of apoptosis of oral epithelial cells and the nutritional requirements of gastric cancer patients after laparoscopic gastroectomy with D2 lymph node excision.

The manuscript submitted for review requires correction.

As it was not submitted using The Microsoft Word or the LaTeX template that would allow to indicate the lines to be corrected, comments were placed directly in the manuscript. Please find attached file.

Reviewer 2 Report

The paper submitted by Gao C. et al. is a prospective cohort study aimed to validate the predictive value of oral epithelial cell apoptosis rate in perioperative nutritional needs and to suggest a new clinical nutritional strategy for gastric cancer patients who had undergone surgery.

The authors explained how patients were selected and applied standard statistical techniques to demonstrate their thesis.

The paper seems to represent a novelty in the field of gastric cancer and has its importance precisely in suggesting how a non-invasive technique (oral epithelial cell apoptosis rate) can be used to represent the patients' perioperative nutritional needs: a proper nutrition strategy will held in a better recovery of cancer patients.

MAJOR REVISION:

This manuscript is well written but it has a great weakness that is the small sample size available for the analysis. The small sample size (53 patients) has been broken into two smaller sub-groups, further reducing the number of individuals and the statistical power of the results. The authors themselves admit this problem.

In view of this consideration, I strongly suggest the author to estimate and control the suitable sample size and the margin of error of their analysis. This kind of statistical test should make their conclusions statistically strong. Without these estimates their result has to be considered only indicative.

MINOR REVISIONS:

Page 5

Paragraph Allocation of patients

It is not clear the criteria of sample subdivision in formular nutrition (FN) and inflection point nutrition (IPN). Here or elsewhere in the text this has to be explained.

Figure 1

I have not understood  the calculation. You started from 169 individuals and you remain with 53 patients (27 IPN and 26 FN). This means hat 116 patients were excluded. However, in the first box you excluded: 31+42+53=126 patients. Why this sum is not equal to 116? Please explain this.

Moreover, at the end of the figure you have 27 IPN patients and 27 FN patients. FN patients should be 26 as stated in the text. Please control this figure carefully.

Page 9

Table 1.

Please, put Gender as the other categorical variable in the table as statistical test are the same. Let the continuous variable together and he categorical too.

In the legend, please specify that values of means+/-standard deviation are shown.

Usually, in table, significant p-value are indicated with asterisks as superscripts and not in bold.

Page 11

At the end of the first paragraph you have to correct if you are referring to Figure 2 or Figure 3 or both. How you stated, it is not clear.

Figure 3

Please insert in a legend the meaning of uppercase A, B,C, D, E, F.

From this figures is not so easy to see what the authors explain in the text and seems to be only an indicative results. Please improve the explanation with further analysis.

Page 14

Inflection-point nutrition supply and postoperative recovery

You stated that after surgery the median time to first flatus is  for IFN group equal to 68 hours (IQR: 48, 80) but this are not the data reported in Table 4. Please correct the text or the table.

In view of these considerations, I suggest the author to estimate the suitable sample size and the margin of error of their analysis. This further analysis has to be discussed in the future manuscript. Moreover a carefully reading of the paper is suggested to avoid errors.

In my opinion, now the manuscript cannot be published. However, If the author can perform the requested preliminary analysis and the results are positive, the paper can be reconsidered and may be suitable to publication as I believe that the idea underling their analysis can be innovative.

Reviewer 3 Report

The work presented is well designed
my suggestions are
put the manuscript in the journal format
table 1 is very large, you can divide it into two tables
the quality of the figures is low, they can make the figures professionally.

Author Response

thank you for the comment. We have revised the manuscript. Please update with the revised manuscript.

Reviewer 4 Report

For the first time the authors described the inflection-point phenomenon and applied it in a novel clinical nutrition therapy strategy. Univariate analysis revealed that age, BMI≤24Kg/m2, serum albumin<35g/L, operation time ≤300mins and average energy intake≥25Kcal/Kg/day were significantly associated with infection phenomenon in gastric cancer patients after laparoscopic D2 gastrectomy.

 Multivariate analysis confirmed that age, operation time ≤300 min and average energy intake≥25Kcal/Kg/day were independent predictive factors of inflection phenomenon.

In this study, we developed a novel nutrition therapy (Inflection-point nutrition) based on the apoptosis rate of patients’ mucosa epithelial cells. Accurate nutrition assessment is essential for effective nutrition support therapy. In our previous study, we found that the apoptosis/proliferation ratio (A/P) of exfoliated oral mucosal epithelial cells (EOMECs) varied and associated with the change of body nutritional status

The rate of both apoptosis and proliferation was reduced in patients with malnutrition. Oral epithelial cells were cells with active renewal and strong proliferative state. The apoptosis and proliferative rate of the oral epithelial cell can be affected by the change of nutrition status of the body. Some animal study found that malnutrition leads to lower apoptosis rate while others indicating that malnutrition is associated with a significant increase of spontaneously apoptotic cells in the thymus and spleen

By active monitoring dynamics of patients’ mucosal epithelia cell apoptosis, we found that the apoptosis rate elevated after nutrition support therapy, reflecting the efficacy of nutrition support therapy.

Round 2

Reviewer 2 Report

I have reconsidered the revised paper by Gao et al. The authors have answered to most of the requests I have submitted to their paper. Now the paper is easier to read and the data and results are increased in clarity. The problem of the small sample still remains but the authors have underlined more times that their result are only indicative.

In view of these considerations, I think that the paper can be published